# Not the Cat’s Meow? The Impact of Posing with Cats on Female Perceptions of Male Dateability

**DOI:** 10.3390/ani10061007

**Published:** 2020-06-09

**Authors:** Lori Kogan, Shelly Volsche

**Affiliations:** 1College of Veterinary Medicine and Biomedical Sciences, Colorado State University, Fort Collins, CO 80523-1601, USA; lori.kogan@colostate.edu; 2Department of Anthropology, Boise State University, Boise, ID 83725, USA

**Keywords:** dating, cats, personality, sex roles, human–animal interactions

## Abstract

**Simple Summary:**

People use dating sites to look for both long-term and short-term potential partners. Previous research suggests that the presence of a pet may add to women’s perceptions of male attractiveness and dateability. This study sought to understand to what degree, if any, the presence of a cat has on women’s perceptions of men. Women responded to an online survey and rated photos of men alone and men holding cats on measures of masculinity and personality. Men holding cats were viewed as less masculine; more neurotic, agreeable, and open; and less dateable. These results varied slightly depending whether the women self-identified as a “dog person” or a “cat person.” This study suggests that a closer look at the effects of different companion species on perceived masculinity and dateability is warranted.

**Abstract:**

The aim of this study was to investigate whether men were considered more attractive when posing for a photo alone or holding a cat. Prior research suggests that women view pet owners as more attractive and dateable than non-pet owners; however, this effect was strongest with dog owners. We hypothesized that men posing with cats would be more attractive than those posing alone. Using an online survey, women viewed images of a man posing alone or with a cat and rated the men on the Bem Sex Role Inventory (BSRI) and the Big Five Inventory. Women viewed men as less masculine when holding the cat; higher in neuroticism, agreeableness, and openness; and less dateable. These findings suggest that pets continue to play a role in women’s mate choices and dating preferences, but that a closer look at the effects of different species of pets is warranted.

## 1. Introduction

People use dating sites for various reasons. Some use the sites to find a long-term partner while others are looking for a casual, short-term encounter. Depending on their reason for using online dating platforms, the qualities and traits each person seeks in a partner may vary. For example, an individual looking for a long-term relationship may want to find someone caring and responsible, while someone seeking a casual, short-term relationship may focus more on physical attractiveness. Buss [1] identified these distinctions as long-term and short-term mating strategies, and argued that, while women may initially seek a physically masculine partner (“good genes”) for short-term mating, they still often evaluate that partner as a potential long-term mate. Research since the early 1990s has further refined this understanding to emphasize the influence of key factors such as: (1) the timing of a woman’s ovulatory cycle; (2) age and life history; (3) current relationship status; and (4) cultural factors defining what is “manly” [2]. However, it is worth noting that, despite these variables, women under the age of 30 seem to consistently prefer men who are higher in physical and behavioral masculinity.

Fiore and colleagues’ [3] research on attractiveness in online dating profiles supports the connection between ratings of masculinity and women’s positive perceptions of profiled men as “attractive.” When participants were asked to rate online profiles that included both photos and free text descriptors, high masculinity ratings were the key predictor of overall attractiveness. Likewise, men’s attractiveness was connected to perceptions of photos as trustworthy and extraverted. 

Mitchell and Ellis’ research [4] may help inform how pets in men’s dating profile photos may influence the viewer’s perception of the man as masculine or feminine, and even perceptions regarding sexual orientation. A college sample of 485 individuals, aged 18–23, watched short videos of two men playing a popular board game and then answered a series of questions regarding the men in terms of labels, sex roles, and personality traits. The authors found that when labeled a “dog person” the men were perceived as more masculine than when labeled a “cat person.” This is congruent with Perrine and Osbourne’s [5] findings that self-identified or externally labeled “dog persons” were perceived as more masculine and independent. 

As relationships between humans and companion animals continue to evolve, pets are becoming increasingly important in the lives of their owners. A growing body of literature suggests that many people perceive their pets as members of the family, up to and including the role of child [6,7,8]. Because of this, pets are often relevant in the formation and development of new romantic relationships. Gray et al. [9] found that women consider how potential partners behave around pets when determining whether to pursue or continue a relationship. They also found that women self-report more attraction to pet owners compared to non-owners. This hypothesis was supported by Gueguen and Ciccoti’s study [10] that found men were more successful in obtaining women’s phone numbers when accompanied by a dog (28.3%) than when they approached women alone (9.2%).

When comparing dog and cat owners, dog owners have been rated as having more potential as a mate than cat owners [9]. This may be due to perceived differences between dogs and cats as companions. Dogs are typically recognized as being more social than cats, and thus requiring more attention and direct care [11,12,13,14]. Cats are perceived as more solitary and may interact less directly or less frequently with humans. While both species require care, perhaps being viewed with a dog suggests a greater ability to connect and care for another being. 

Because the association with a pet appears to increase a person’s rated attractiveness, we hypothesized that men posing with cats would be considered more attractive and desirable for short-term causal dating than when posing alone. In addition to attractiveness, we predicted that the cat’s presence would make the men seem more trustworthy, gentle, and caring, alluding to the possibility of a potentially valuable long-term mate and future father. However, we recognize the possibility that a women’s identification as “dog person” or “cat person” may influence their ratings. To this end, we hypothesized that women who identify as a “dog person” may be less likely to perceive a “cat person” as datable. 

## 2. Materials and Methods

To conduct this study, an online, anonymous, cross-sectional survey was developed using Qualtrics (Qualtrics, Inc., Provo, UT, USA). The survey was designed, reviewed, and tested by the co-investigators of the study and their colleagues, and a pilot tested by eight individuals for ambiguity and/or potentially missing or inappropriate response options. Revisions were made based on the results of the pilot testing. The final survey and study design were approved by the Colorado State University Institutional Review Board (IRB # 332 −18H). Survey respondents were recruited in January 2020 through Amazon’s Mechanical Turk (MTurk; Amazon Inc., Seattle, WA, USA) platform, an open online marketplace providing affordable access to potential survey respondents (Buhrmester, 2011). Survey respondents receive small monetary compensation for completing the surveys (e.g., ≤50 cents per survey). The diversity of participants recruited through MTurk is greater than typical Internet samples and the quality of data collected meets acceptable psychometric standards for published social science research (Buhrmester, 2011). In order to minimize the influence of geographic and cultural differences on respondent data, the survey was made available only to people who identified as female, heterosexual, and aged 18–24 who reside in the United States. 

Two versions of the survey were created, with each version featuring pictures of one male so that each man could serve as his own control (see Figure 1 for images used). All participants were randomly assigned to take Survey 1 or Survey 2. Male 1 was 20 years of age and Male 2 was 21 years old. In each survey, participants were shown two different pictures of either Male 1 or Male 2. All pictures were taken with the males sitting in a chair with a white background. They each wore a blue button-down shirt and jeans. In one picture, the males were pictured alone and in the other picture, they were each photographed with a cat in their lap. The picture orders (alone first or cat first) were randomized. 

After a picture was displayed, the participants were asked to rate the men on several attributes, including perceived personality (BFI-10, Big Five Inventory, short version), perceived masculinity or femininity (12-Item Bem Sex Role Inventory [BSRI-12]), and perceived dateability, asking directly if each participant would consider dating the man in the photo for a short or long term. Finally, general demographic data were collected, including a verification of the participant’s age, which included the options “under 18”, “between 18 and 24”, and “over 24”. All participants who reported ages outside the target range (18–24 years) were removed from the sample before analysis.

### 2.1. BFI-10 (Big Five Inventory- Short Version) 

After a picture was displayed, the participants were asked to rate the man on several attributes. These attributes included the 11 items that comprise the BFI-10 (Big Five Inventory, short version) to assess the big five personality dimensions (Extraversion, Agreeableness, Conscientiousness, Neuroticism, and Openness). These attributes include the following: Is reserved;Is generally trusting;Tends to be lazy;Is relaxed, handles stress well;Has few artistic interests;Is outgoing, sociable;Tends to find fault with others;Does a thorough job;Gets nervous easily;Has an active imagination;Is considerate and kind to almost everyone.

For each of these traits, participants were asked to look at the picture and rate how well they agree that the statements described the pictured male’s personality from strongly disagree (1) to strongly agree (5). To assess each of the five personality dimensions, the mean for the two questions that pertain to that dimension (Agreeableness has 3 items) was calculated and compared to a comparison sample provided by John and Srivastava (1999) who reported means for each trait by years of age (ages 21–60). Since our sample was composed of women aged 18–24, and the males in the pictures were 20 and 21 years of age, we selected 21 years of age to compare our scores.

### 2.2. 12-Item Bem Sex Role Inventory (BSRI-12)

The 12-Item Bem Sex Role Inventory (BSRI-12) was used to assess how participants viewed the depicted men with and without the cat in terms of sex roles. They were given a list of characteristics and asked to rate how well the attributes describe the depicted male using a 7-point Likert scale, anchored with 1 (not applicable/does not describe at all) to 7 (very much/totally applies). Respondents responses were calculated to create both a femininity and masculinity score. 

The 12 traits included:Defends own beliefs;Strong personality;Has leadership abilities;Makes decisions easily;Dominant;Acts as a leader;Affectionate;Sympathetic;Sensitive to needs of others;Warm;Tender;Gentle.

### 2.3. Dateability

Participants were asked if they consider themselves a “dog person”, “cat person”, “both a dog and cat person”, or “neither a dog nor cat person”. A question about current relationship status was included in this section: “Are you currently in a relationship?”, with yes or no options. If they answered no, participants were asked (after viewing each picture): “How likely would you be to ‘swipe right’ (select for a casual date/encounter)?”. Options included: “Would never consider it”, “Maybe, but not likely”, “Perhaps”, “Yes, likely”, “Absolutely yes”, and “I don’t casually date (hook-up)”. If they answered yes, the dating question was modified to: “If you were not in a relationship right now, how likely would you be to ‘swipe right’ (select for a casual date/encounter)?”. This same technique was used for the long-term relationship dating question (“If you were looking to start a long-term relationship, how likely would you be to go on a date with this man?”). The response items in this instance were the same, except for the last option. Instead of the last option being “I don’t casually date (hook-up)”, it was “I am not interested in dating for a potential long-term relationship”.

## 3. Results

### 3.1. Survey/Male 1 Participant Demographics

A total of 1385 people responded to the survey. Of those respondents, 708 met the criteria of being female and between 18 and 24 years of age. From those 708, the largest percentage indicated they were a dog person (337, 47.6%), white (443, 62.7%), and had a bachelor’s degree (323, 45.6%) (Table 1.). The remaining 677 responses were either automatically terminated by MTurk for not meeting the inclusion criteria set in the survey parameters or removed manually for being incomplete surveys.

The 11 items of the Big Five Inventory were used to calculate the mean scores for the depicted male with and without a cat for each of the five traits. The means and standard deviations for each trait are shown in Table 2. When Male 1 was viewed in the picture by himself, he was perceived as more extraverted then when pictured with the cat. When he was pictured with the cat, he was perceived as more agreeable, neurotic, and open then when pictured alone. 

The 12 item Bem Sex Scale was used to assess the perception of the depicted male’s masculinity and femininity by the participants (Bem, 1974). The median feminine scale scores for the two picture conditions (alone (31) vs. with cat (34)) were compared via a paired *t*-test and found to be statistically different (*t* = −7.13 (df 707), *p* < 0.001) with the cat condition viewed as more feminine. Similarly, the median masculine scores for the two conditions (alone: 28, with cat: 26) were compared with a paired *t*-test and found to be statistically different (*t* = 6.76 (df 707), *p* < 0.001) with the alone condition viewed as more masculine. 

### 3.2. Datability

Participants were asked to rate the two pictures of Male 1 in terms of how likely they would be to consider him for short-term/casual dating or a long-term dating/relationship. Participants that indicated they would never consider short-term or long-term dating were excluded from the analysis. A related samples Wilcoxon signed rank test was used to test for significant differences between the two conditions. There was a significant difference between perceptions of the man when viewed alone versus with a cat for short-term dating (n = 656), X = −4.88 (*p* < 0.001), with the alone condition viewed more favorably. Comparing the cat picture with the alone picture yielded 329 ties, 206 negative differences (less likely to casually date the male in the cat condition) and 121 positive differences (more likely to casually date the male in the cat condition).

Similarly, a related samples Wilcoxon signed rank test was used to determine whether there was a difference between the two conditions (alone vs. cat) for long-term dating/relationship. There was a significant difference between the conditions (n = 690), X = −4.62 (*p* < 0.001). When comparing the picture of the man with the cat vs. alone, there were 351 ties, 214 negative differences (cat picture viewed as less favorable for a long-term relationship), and 125 positive differences (cat picture viewed as more favorable for a long-term relationship).

To assess how the pet status of the participants (dog person, cat person, both, neither) impacted the perceived differences between the pictures with the man alone and with the cat, the delta for each participant’s responses for the alone compared to the cat picture were calculated. A Chi Square was used to assess significant differences between the four groups of pet status on participants’ stated willingness to date long-term. A significant difference was found between the groups (Chi Square = 40.53 (df 21), *p* = 0.006), whereby dog people were less likely to favor the picture with the cat while cat people were more likely to favor the cat picture. Participants in the “both” and “neither” categories were more likely to favor the male when pictured alone (Table 3.).

The same analysis was done to assess significant differences between the four groups of pet status on participants’ stated willingness to short-term/casual date. Results were not significantly different (Chi Square = 26.65 (df 21), *p* = 0.183). Participants falling into the “dog person” and “neither” categories were less likely to favor the picture with the cat, while participants in the “cat person” and “both” categories did not favor one condition over the other (Table 3.).

### 3.3. Survey/Male 2 Participant Demographics

A total of 1404 people responded to this survey. Of those participants, 680 met the criteria of being female and between 18 and 24 years of age. Out of those 680, the largest percentage, indicated they were a dog person (305, 44.9%), white (425, 62.5%), and had a bachelor’s degree (302, 44.4%). These results are indicated in Table 4. The remaining 724 responses were either automatically terminated by MTurk for not meeting the inclusion criteria set in the survey parameters or removed manually for being incomplete surveys.

As with Survey/Male 1, the mean scores for each picture of Male 2 were calculated using the 11 items of the Big Five Inventory. This was done for each of the five traits and is shown in Table 5.

Using the 12 item Bem Sex Scale (Bem, 1974), perceived masculinity and femininity levels were determined for each picture condition. For the feminine scale, the two medians scores (alone: 30, with cat: 32) were compared using paired t-test and found to be statistically different (−7.40 (667), *p* < 0.001). The masculine median scores (alone: 26, with cat: 25) were also found to be statistically different (2.32 (670), *p* = 0.020).

### 3.4. Datability

Related samples Wilcoxon signed rank tests were used to determine a difference between participants’ rating of the male with and without a cat in terms of short-term/casual dating and long-term dating/relationship. The participants who indicated they would never consider short-term or long-term dating were excluded from the analyses. For short-term dating, the related samples Wilcoxon signed rank test (n = 618), X = −1.99 (*p* = 0.046) revealed that, when comparing the cat picture with the alone picture, there were 354 ties, 143 negative differences (in which the cat picture was viewed as less favorable), and 121 positive differences (in which the cat picture was viewed as more favorable). 

For long-term dating/relationship perceptions, related samples Wilcoxon signed rank tests found no significant difference between the picture of the man alone and with the cat (n = 655), X = −1.84 (*p* = 0.066). When comparing the cat picture with the alone picture, there were 383 ties, 142 negative differences (whereby the cat picture viewed as less favorable), and 130 positive differences (whereby the cat picture viewed as more favorable) (Table 6). 

In assessing the impact of pet status, Chi Square tests revealed a significant difference (46.31 (24), *p* = 0.004) between the four groups of pet status on participants’ perceptions related to willingness to short-term/casual date. Participants self-identified as a “dog person” and “neither” were less likely to favor the picture with the cat, while participants who identified as a “cat person” and “both” did not favor one condition over the other. Chi Square tests resulted in significant differences (Chi Square = 51.05 (df 21), *p* < 0.001) between the groups in terms of long-term dating/relationship. Participants in the “dog person” and “neither” categories were less likely to favor the picture with the cat while participants in the “cat person” and “both” categories were more likely to prefer the male in the cat picture (Table 6.).

## 4. Discussion

This study found that college-age women viewing a photo of a man alone versus a photo of the same man holding a cat rated the man holding the cat as less masculine; higher on neuroticism, agreeableness, and openness; and ultimately, less datable in the short or long term. Yet, it is important to note that these findings were influenced by whether the female viewer self-identified as a “dog” or “cat” person, suggesting that American culture has distinguished “cat men” as less masculine, perhaps creating a cultural preference for “dog men” among most heterosexual women in the studied age group.

Prior research [1,2] suggests that women desire different traits in a partner for short-term mating strategies (i.e., hook-ups) compared to long-term mating strategies (i.e., committed relationship). For short-term relationships, women are more likely to seek a man high in physical masculinity (i.e., large chins, certain facial features) and behavioral masculinity (i.e., dominance display). Part of females’ short-term mating strategies also includes evaluating the potential mate for suitability as a long-term partner. In the case of our study, women viewed the photo of the man alone as more masculine and more datable for both short-term and long-term pairing. This supports the hypotheses that women are more likely to seek masculinity first, then consider other components of the potential mate (i.e., perceived personality, suitability for long-term relationship). 

Because Gray and colleagues [9] found that women evaluated potential dates positively based upon ownership of a pet, we hypothesized that the photo of a man holding a cat would be deemed more dateable. Contrary to this hypothesis, only female respondents who identified as a “cat person” found him more desirable for short-term or long-term relationships. Gosling, Sandy, and Potter [15] found that self-identified “dog people” and “cat people” differ in personality markers. In their study, over 4000 participants completed the Big Five Inventory and self-identified as a “dog person”, “cat person”, both, or neither. The analysis found that self-identified “dog people” were higher on extraversion, agreeableness, and conscientiousness, while “cat people” were higher on neuroticism and openness. Our participants echo this cultural belief. The women who participated in our survey identified the man as more neurotic and open when holding the cat, and more extraverted when viewed alone. It is possible that the women completing our survey unconsciously defaulted to the idea that the man alone was a “dog person” or, perhaps, more willing to date a “dog person”. Interestingly, the presence of the cat did enhance agreeableness, perhaps suggesting that agreeableness is a trait present in the identity “pet owner” and alluding to why Gray and colleagues [9] found women were more interested in dating pet owners than non-owners, regardless of pet type.

It is interesting to note that, while Male 1 and Male 2 were not rated by the same respondents, the significant differences in the Big Five ratings for Male 1 (increased Extraversion when alone, increased Agreeableness, Neuroticism, and Openness when posing with the cat) were not found in Male 2. Because these men were not rated by the same samples, it is difficult to say if this is the result of variation in the sampled women or some perception of the men themselves. While attempts were made to control variables of individual dating preferences, these confounding variables are inherently present in any sort of comparative survey, contributing to statistical noise in the results. Despite the discrepancy in the Big Five ratings, it is relevant that in both instances, the photos of the men alone were perceived as more masculine and the photos of the men holding cats were less likely to be perceived as dateable, except by “cat people”.

The current study used a short form of the Bem Sex Role Inventory (BSRI) (BEM 1974). The BSI had been used previously to determine whether self-labeling or external labeling as a dog or cat person is related to one’s self-reported personality as masculine, feminine, independent, athletic, or dominant. Surveying 126 undergraduates, Perrine and Osbourne [5] found that self-identified or externally labeled “dog persons” were perceived as more masculine, regardless of the respondent’s gender, experience with pets, or self-label. This effect did appear stronger for male “dog persons”. Being labeled “cat person” did not interact significantly with perceptions of femininity or masculinity. They did, however, find that “dog people” rated significantly higher on independence than “cat people”. It is possible that our respondents also viewed the photo of the man with the cat as less independent and therefore less attractive. 

Given that our respondents perceived the photo of the man holding a cat as more feminine, it is also possible they subconsciously (or consciously) perceived him as being gay. Kranz, Pröbstle, and Evidis [16] found that homosexual men were rated as more feminine and less or equally masculine than heterosexual men as rated by college aged men and women. If this indeed remains a cultural perception, then women who perceive a man holding a cat as less masculine could, potentially, believe he is also gay. This may be compounded by the perceptions of him as more neurotic, as well. However, additional research would be needed to confirm this possibility.

This supports our first finding, that our respondents perceived a man holding a cat as less masculine and less datable for both short-term encounters and long-term relationships. This returns us to Buss’s [1] assertion that women prefer men with “good genes”, often defined as more masculine traits. Clearly, the presence of a cat diminishes that perception.

## 5. Limitations

As with any research, our study has certain limitations. Most notably, this is a specific target age range (18–24 years) of heterosexual women living in the United States who are also well-educated. Hence, we would not expect these findings to generalize without first testing them with a new sample and a new pair of photos with an appropriately aged man. It may also be relevant to consider different archetypes of men (i.e., preppy, “bad boy”, etc.) given the role culture and identity play in mate preferences [2]. However, this is a methodological challenge in any survey. Since each male served as his own control (i.e., posing alone vs. with cat), it is most likely the measured differences are related to the presence of the cat rather than the respondents’ personal preferences.

Likewise, the men in the photos, and a large majority of our sample, are white or white-passing. As such, we cannot be certain that our findings generalize to other racial or ethnic categories. The same could also be said for potential respondents’ perceptions of socioeconomic status, given the bias toward well-educated women in our sample. In fact, a surface level analysis of our own data suggests that the presence of cats may only be relevant to those individuals seeking to date a white male. The more middle-class appearance (i.e., shorter hair, button down shirt) of the men in the photos may also be a limitation for respondents who may prefer another archetype or personality for dating. Perceptions of masculinity may or may not vary across ethnic and racial boundaries or socioeconomic status, and the mediating role pets play on these perceptions has yet to be investigated fully. 

It is also worth noting that in using the Bem Sex Role Inventory, the measurements of masculine and feminine traits need reflection. The validity and use of this scale remain open to debate among sex and gender researchers. Likewise, sex and gender, while relatively consistent in the evolutionary psychology literature, are considerably responsive to cultural norms and performance. Despite this, we chose to use the BSRI because we are unaware of a more validated or preferred scale for this purpose. The future development of a replacement scale is also outside the scope of the current study objectives.

Finally, though we chose the study design to control for confounding variables, its simplicity may have made our research objective too obvious to respondents. We attempted to mitigate this possibility by randomizing which photo respondents saw first (alone or with cat). Regardless, given the educational status of our sample, we cannot rule out the potential that some of the respondents were biased by guesses as to the purpose of the study.

A follow-up study with a third comparative photo of a man with a dog could serve to further test the impact of pets by looking across species. For example, would women find the man alone more or less masculine than a man with a dog? Further, what role would the size/breed of the dog play in these perceptions? Attempting to address these questions, while repeating research to control for the limitations in the current study, could provide a rich avenue of inquiry. An understanding of women’s mating preferences in relation to the presence and type of pet may also lend itself to qualitative analysis not included here. These questions, and more, remain an area for future research.

## Figures and Tables

**Figure 1 animals-10-01007-f001:**
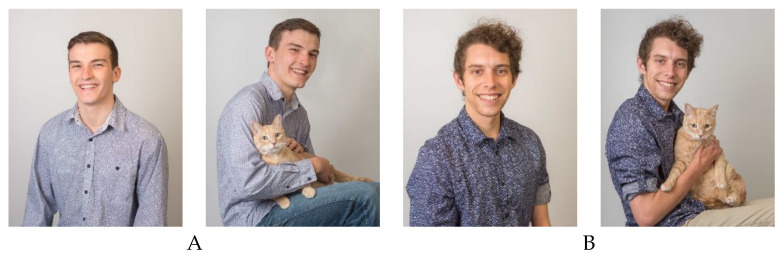
Photos presented to survey respondents. **A**. Male 1 alone and with cat. **B**. Male 2 alone and with cat.

**Table 1 animals-10-01007-t001:** Demographics of participants: Survey/Male 1.

**Pet Type**	***N***	**%**
Dog Person	337	47.6
Cat Person	135	19.1
Both	168	23.7
Neither	68	9.6
**Race/Ethnicity**	***N***	**%**
White	443	62.7
Black or African American	97	13.7
Hispanic or Latino	68	9.6
Asian	66	9.3
Native American or Alaskan Native	11	1.6
Native Hawaiian or Pacific Islander	2	0.3
Other	6	0.8
NA/Prefer to Not Say	14	2.0
**Education**	***N***	**%**
Did not Graduate High School	3	0.4
High School	131	18.5
Technical or Vocational Degree	47	6.6
Associate’s Degree	117	16.5
Bachelor’s Degree	323	45.6
Master’s Degree	73	10.3
Doctoral or Professional Degree	14	2.0

**Table 2 animals-10-01007-t002:** Survey/Male 1 Big Five Inventory.

	Extraversion	Agreeableness	Conscientiousness	Neuroticism	Openness
Comparison Alone vs. Cat	3.25 (0.90)	3.64 (0.72)	3.45 (0.73)	3.32 (0.82)	3.92 (0.66)
Male 1 Alone	3.32 (0.80)	3.59 (0.67)	3.53 (0.76)	2.71 (0.75)	3.11 (0.66)
Male 1 Cat	2.96 (0.81)	3.67 (0.70)	3.50 (0.74)	2.83 (0.78)	3.25 (0.70)
Paired *t*-Tests	*t* = 10.07 (707), *p* < 001	*t* = −3.39 (707), *p* = 0.001	*t* = −.98 (707), *p* = 0.329	*t* = −3.81 (707), *p* < 0.001	*t* = −4.92 (707), *p* < 0.001

**Table 3 animals-10-01007-t003:** Male 1 Dateability.

		Would Never Consider It	Maybe, But Not Likely	Perhaps	Yes, Likely	Absolutely Yes	Don’t Casually Date/Not Interested in Dating
Male Alone	Casual dating	63 (8.9%)	136 (19.3%)	197 (27.9%)	206 (29.2%)	65 (9.2%)	38 (5.4%)
	Long-term relationship	68 (9.6%)	136 (19.3%)	227 (31.2%)	182 (25.8%)	78 (11.0%)	15 (2.1%)
Male with Cat	Casual dating	101 (14.3%)	143 (20.3%)	189 (26.8%)	178 (25.3%)	55 (7.8%)	38 (5.4%)
	Long-term relationship	103 (14.5%)	161 (22.7%)	205 (29.0%)	163 (23.0%)	68 (9.6%)	8 (1.1%)

**Table 4 animals-10-01007-t004:** Demographics of participants: Survey/Male 2.

**Pet Type**	***N***	**%**
Dog Person	305	44.9
Cat Person	146	21.5
Both	162	23.8
Neither	67	9.9
**Race/Ethnicity**	***N***	**%**
White	425	62.5
Black or African American	100	14.7
Hispanic or Latino	75	11
Asian	49	7.2
Native American or Alaskan Native	6	0.9
Native Hawaiian or Pacific Islander	2	0.3
Other	4	0.6
NA/Prefer to Not Say	19	2.8
**Education**	***N***	**%**
Did not Graduate High School	6	0.9
High School	140	20.6
Technical or Vocational Degree	38	5.6
Associate’s Degree	120	17.6
Bachelor’s Degree	302	44.4
Master’s Degree	63	9.3
Doctoral or Professional Degree	11	1.6

**Table 5 animals-10-01007-t005:** Survey/Male 2 Big Five Inventory.

	Extraversion	Agreeableness	Conscientiousness	Neuroticism	Openness
Comparison Alone vs. Cat	3.25 (0.90)	3.64 (0.72)	3.45 (0.73)	3.32 (0.82)	3.92 (0.66)
Male 2 Alone	3.11 (0.80)	3.55 (0.66)	3.35 (0.77)	2.90 (0.74)	3.24 (0.68)
Male 2 Cat	2.92 (0.81)	3.57 (0.69)	3.33 (0.74)	2.87 (0.75)	3.28 (0.73)

**Table 6 animals-10-01007-t006:** Survey/Male 2 Dateability.

		Would Never Consider It	Maybe, But Not Likely	Perhaps	Yes, Likely	Absolutely Yes	Don’t Casually Date/Not Interested in Dating
Male Alone	Casual dating	117 (17.4%)	154 (22.9%)	166 (24.7%)	141 (21.0%)	45 (6.7%)	50 (7.4%)
	Long term relationship	122 (18.0%)	158 (23.3%)	195 (28.7%)	133 (19.6%)	53 (7.8%)	18 (2.7%)
Male with Cat	Casual dating	137 (20.2%)	165 (24.4%)	149 (22.0%)	132 (19.5%)	53 (7.8%)	41 (6.1%)
	Long term relationship	141 (20.8%)	162 (23.9%)	183 (27.0%)	130 (19.1%)	49 (7.2%)	14 (2.1%)

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
