# Peer review of "Not the Cat’s Meow? The Impact of Posing with Cats on Female Perceptions of Male Dateability"

_animals, 2020, doi:10.3390/ani10061007_

Round 1

Reviewer 1 Report

Rev.  Not the cat's meow

In this paper young women rate a pictured man with a cat and the same man without a cat. Two versions of the survey were conducted, each featuring a different young man in the two photos. The women rated the man in each of the two pictures on his personality, his masculinity/femininity, and dateability.

Using Mechanical Turk, the survey was made available to people identified as female, heterosexual, and aged 18-24. Nonetheless, in both surveys, about half of the participants' surveys were disqualified as not meeting those same criteria. Some explanation needs to be provided as to why half of the surveys were not eligible for inclusion.

In Survey 1 significant differences were found for all of the Male Big 5 except one, conscientiousness. In Survey 2 no significant differences were found for the Male Big 5. There was no discussion of these contrasting results, and no attempt to integrate or explain them. The significant results were mentioned in the abstract but not the non-significant ones.

The photos of the men that were used should be attached.

It is quite a simplistic study design--rating a single man with and also without a cat. A lot is resting on the appearance of this one person. Plus the raters seemingly could guess the nature of the hypotheses as relating to the presence of the cat. These are limitations of the study design that need to be mentioned.

Typos

LInes 99, 245, 298, 303, 310

Author Response

Reviewer 1

  1. Typos:
    1. Line 99: changed “along” to “alone.”
    2. Line 245 (now line 249): changed “significantly” to “significant”
    3. Line 298 (now line 312-313): changed “women who perceived a man … lead them to believe” to “women who perceive a man holding a cat as less masculine could, potentially, believe he is also gay.”
    4. Line 303 (now line 317): removed “who”
    5. Line 310 (now line 324): changed “identify” to “identity”
  2. Explanation of individuals not eligible for inclusion: We added a line to both Male 1 and Male 2 explaining that MTurk either automatically removed individuals who did not meet the stated inclusion criteria (18-24 years of age and female) or were manually removed for not completing the survey.
  3. Discussion of contrasting or non-significant results: We actively chose not to compare the two men to each other in our analysis because they were rated by completely different sample pools. However, we thank the reviewer for drawing our attention to this interesting disparity and we added a paragraph to the Discussion contemplating how this may be a variation in the respondents or representative of a variation in the men themselves. Also noted that, despite the Big Five variance, the dateability and masculinity findings were consistent.
  4. Inclusion of photos: Figure 1 inserted into Methods and Materials after line 100.
  5. Additional Limitations:
    1. Individual appearance of men/rater’s preferences: We address the importance of considering different types of men (lines 312-316), explaining that this is why each man served as his own control.
    2. Respondents guessing our hypothesis/project purpose: We added the reviewer’s suggestion related to the potential for respondents guessing our hypothesis, and included a reminder that we attempted to control for this issue by randomizing the order in which respondents were presented with the photos in each survey.

Reviewer 2 Report

I found this paper fascinating to read. The study design is original. The methods are sound. I have three suggestions for strengthening the paper. 

In a book titled The Photographed Cat: Picturing Human-Feline Ties, 1890-1940, Arnold Arluke and Lauren Rolfe have a chapter on gender and photos of cats and people. They point out that the gendering of dogs and cats existed even in the early twentieth century. Arluke and Rolfe found a specific set of codes for communicating the symbolic function of gender and cats. I would recommend incorporating some of their claims, as they would provide a valuable and intriguing historical grounding for the contemporary findings. 

In Section 5, Limitations, along with age, gender, and sexual preference, add that the sample is more highly educated than the general American public. Current rate for women (receiving BA) is just under 37%. I don't know that more education influences preference of dog- or cat-person for a mate, but it's an important detail about your sample. 

Also in Limitations, I'd like to see a mention of some of the pitfalls with using the Bem Sex Role Inventory, especially for the masculine/feminine binary (it was designed for androgyny). In addition, its validity has been questioned and the results depend on the form and scoring used. 

Author Response

Reviewer 2

  1. Photographed Cat: We thank the reviewer for the suggestion of Arluke and Rolfe’s book. However, given that most university libraries remain closed, accessing this book is difficult (neither author currently own the book). Likewise, making time to read this book before revisions are due may be difficult. We do appreciate the benefit a manuscript would receive from a historical grounding, and were we to expand this research, we would definitely provide more sociohistoric context in future work.
  2. Adding Education to Limitations: We thank the reviewer for drawing our attention to this exclusion, and we have added “well educated” to our list of biases in the sample. This certainly does limit our ability to generalize the work, and as noted, not much has been done regarding education and pet-type or mate-type preference. Perhaps that is our next project!
  3. BSRI Limitations: We added a paragraph to the Limitations discussing the BSRI, also noting that, to our knowledge, there is not currently a more validated scale for considering the perceived masculinity/femininity of an individual. Additionally, sex and gender scales are outside the scope of the project, so we do not address this barrier extensively in the manuscript.